

# *AURKB*: a promising biomarker in clear cell renal cell carcinoma

Bangbei Wan[1,*], Yuan Huang[2,*], Bo Liu[3], Likui Lu[4] and Cai Lv[1]

[1] Urology, Haikou Municipal People's Hospital and Central South University Xiangya Medical College Affiliated Hospital, Haikou, China
[2] Neurology, Haikou Municipal People's Hospital and Central South University Xiangya Medical College Affiliated Hospital, Haikou, China
[3] Laboratory of Developmental Cell Biology and Disease, School of Ophthalmology and Optometry and Eye Hospital, Wenzhou, China
[4] Institute for Fetology, First Affiliated Hospital of Soochow University, Suzhou, China
[*] These authors contributed equally to this work.

## ABSTRACT

**Background**. Aurora kinase B (*AURKB*) is an important carcinogenic factor in various tumors, while its role in clear cell renal cell carcinoma (ccRCC) still remains unclear. This study aimed to investigate its prognostic value and mechanism of action in ccRCC.
**Methods**. Gene expression profiles and clinical data of ccRCC patients were downloaded from The Cancer Genome Atlas database. R software was utilized to analyze the expression and prognostic role of *AURKB* in ccRCC. Gene set enrichment analysis (GSEA) was used to analyze *AURKB* related signaling pathways in ccRCC.
**Results**. *AURKB* was expressed at higher levels in ccRCC tissues than normal kidney tissues. Increased *AURKB* expression in ccRCC correlated with high histological grade, pathological stage, T stage, N stage and distant metastasis (M stage). Kaplan-Meier survival analysis suggested that high *AURKB* expression patients had a worse prognosis than patients with low *AURKB* expression levels. Multivariate Cox analysis showed that *AURKB* expression is a prognostic factor of ccRCC. GSEA indicated that genes involved in autoimmune thyroid disease, intestinal immune network for IgA production, antigen processing and presentation, cytokine-cytokine receptor interaction, asthma, etc., were differentially enriched in the *AURKB* high expression phenotype.
**Conclusions**. *AURKB* is a promising biomarker for predicting prognosis of ccRCC patients and a potential therapeutic target. In addition, *AURKB* might regulate progression of ccRCC through modulating intestinal immune network for IgA production and cytokine-cytokine receptor interaction, etc. signaling pathways. However, more research is necessary to validate the findings.

## INTRODUCTION

Renal cell carcinoma (RCC) is one of the most common malignant tumors of the urinary system, and clear cell renal cell carcinoma (ccRCC) is the most common pathological subtype (*Srigley et al., 2013*). Morbidity and mortality of ccRCC are increasing year by year, while the mechanism of ccRCC development still remain unclear (*Dutcher, 2013*).

Corresponding author
Cai Lv, lvcai815@163.com

Hence, biomarkers that can be used to diagnose, treat and predict prognosis of ccRCC, are urgently needed.

Aurora kinase B (*AURKB*), located on human chromosome 17p13.1, encodes a member of the aurora kinase subfamily of serine/threonine kinases. Previous researches have reported that aberrant *AURKB* expression is related to tumorigenesis and progression of tumors (*Zhu et al., 2019*). Single nucleotide polymorphisms (SNPs) of *AURKB* were associated with occurrence of gastric cancer (GC). rs2289590 in *AURKB* might contribute to susceptibility for the development of gastric cancer (*Mesic et al., 2017*). In thyroid cancer, *Sorrentino et al. (2005)* found that *AURKB* was not detected in normal thyroid tissue, but it was overexpressed in thyroid carcinoma. Further experiments indicated that silencing *AURKB* can obviously inhibit the growth of thyroid carcinoma cells. Hence, they thought that *AURKB* was an important protein in the progression of thyroid carcinomas and a promising candidate for targeted treatment. Besides, *AURKB* also plays an important role in non-neoplastic disease. In asthenozoospermia, over-expression of *AURKB* might be associated with development of asthenozoospermia. Over-expression of *AURKB* can decrease glycolytic activities, conferring to the occurrence and progression of asthenozoospermia (*Zhou et al., 2018*). Generally, the several researches have suggested the important role of AURKB in tumors and non-neoplastic disease. However, few studies about the relationship between AURKB and ccRCC have been reported so far and the role of AURKB in ccRCC remains elusive.

In this work, we attempted to reveal the significance of *AURKB* expression in ccRCC and the mechanisms related to ccRCC progression. We compared *AURKB* mRNA expression between tumor tissues and normal tissues. We then analyzed the relationship between *AURKB* mRNA expression and clinical parameters of ccRCC and correlated them with patients'overall survival (OS) and disease-free survival (DFS). Results indicated that patients with high *AURKB* expression have poorer prognosis than patients with low *AURKB* expression. In addition, to further understand the *AURKB*-related biological pathways involved in ccRCC, Gene set enrichment analysis (GSEA) was performed. Results showed that twenty-one genes were evidently enriched in patients with high *AURKB* expression, including intestinal immune network for IgA production, cytokine-cytokine receptor interaction, natural killer cell mediated cytotoxicity, cell cycle and cell adhesion molecules (CAMs), etc.

## MATERIAL AND METHODS

### Database

Gene expression profiles of ccRCC patients and clinical data of patients such as age, gender, pathological stage, histological grade, survival, and outcome were downloaded from The Cancer Genome Atlas (TCGA) database (https://portal.gdc.cancer.gov/). In addition, Drug sensitivity data of ccRCC cell lines were obtained from genomics of drug sensitivity in cancer (GDSC) database (https://portals.broadinstitute.org/ccle/about). We then utilized R software (*R Core Team, 2018*) to process all data.

Firstly, we extracted clinical data of ccRCC patients and data of gene expression profiles. We then obtained clinical data of 530 patients who possessed complete OS information

and a gene expression matrix document. Secondly, we obtained expression of *AURKB* data from the gene expression matrix document and analyzed the relationship between expression of *AURKB* and clinical parameters including age, gender, histological grade, pathological stage, T stage, N stage, and M stage. Thirdly, the ccRCC patients be divided into two groups based on median value of *AURKB* expression (high *AURKB* expression group and low *AURKB* expression group) and analyzed their overall survival (OS) and disease-free survival (DFS). Fourthly, we utilized some clinical parameters, that correlated with prognosis of ccRCC, and *AURKB* to construct a prognostic model. Finally, we analyzed that the difference between sensitivity of AURKB targeted drug and other targeted drugs for ccRCC.

## Gene set enrichment analysis

Gene expression profiles of ccRCC patients were divided into two groups (high expression group and low expression group) according to the median value of expression of *AURKB*. GSEA was utilized to detect potential mechanisms underlying the effect of *AURKB* expression on ccRCC prognosis. Gene set permutations were performed 1,000 times for each analysis. Gene sets with a *p*-value <0.05 and false discovery rate (FDR) <0.05 were regarded as significantly enriched.

## Statistical analysis

All statistical analyses were performed through R software and $p < 0.05$ was regarded as statistically significant. The relationship between expression levels of *AURKB* and clinical parameters was analyzed via the Wilcoxon signed-rank test, Kruskal-Wallis test and logistic regression. The correlation between expression levels of *AURKB,* and patients'OS and DFS were analyzed using the Kaplan–Meier method. Univariate Cox analysis was used to select possible prognostic factors, and multivariate Cox analysis was utilized to verify the correlations between *AURKB* mRNA expression and survival along with other clinical features. A receiver operating characteristic (ROC) curve was used to evaluate the accuracy of models that predicted prognosis using the survival ROC package. An area under the curve (AUC) value of 0.75 or bigger was deemed an excellent predictive value, and values of 0.6 or larger were regarded as acceptable for survival predictions. The chi-square test be used to compare difference between sensitivity of target drugs in ccRCC cell lines.

# RESULTS

## Clinical parameters of patients

The clinical data of 530 ccRCC patients were obtained from the TCGA database, and included age, gender, histological grade, pathological stage, survival, and outcome, etc. (Table 1).

## High *AURKB* expression in ccRCC

Expression levels of *AURKB* in 539 ccRCC and 72 normal kidney tissues were compared via Wilcoxon signed-rank test, and the results showed that *AURKB* was highly expressed in ccRCC compared to normal kidney tissues ($p < 0.05$) (Fig. 1A). We further analyzed

**Table 1  Clinical data of the ccRCC patients.**

| Clinical parameters | Variable | Total (530) | Percentages (%) |
|---|---|---|---|
| Age | <60 | 245 | 46% |
| | ≥60 | 285 | 54% |
| Gender | Female | 186 | 35.56% |
| | Male | 344 | 64.44% |
| Histological grade | G1 | 14 | 2.64% |
| | G2 | 227 | 42.83% |
| | G3 | 206 | 38.86% |
| | G4 | 75 | 14.15% |
| | GX | 5 | 0.94% |
| | Unknow | 3 | 0.56% |
| Pathological stage | Stage I | 265 | 50.00% |
| | Stage II | 57 | 10.75% |
| | Stage III | 123 | 23.20% |
| | Stage IV | 82 | 15.47% |
| | Unknow | 3 | 0.56% |
| T stage | T1 | 271 | 51.21% |
| | T2 | 69 | 12.84% |
| | T3 | 179 | 33.89% |
| | T4 | 11 | 2.04% |
| M stage | M0 | 420 | 79.24% |
| | M1 | 78 | 14.71% |
| | Mx | 30 | 5.66% |
| | Unknow | 2 | 0.37% |
| N stage | N0 | 239 | 45.09% |
| | N1 | 16 | 3.01% |
| | NX | 275 | 51.88% |
| Disease free status | Recurred/Progressed | 125 | 23.58% |
| | Disease free | 308 | 58.11% |
| | Unknow | 97 | 18.30% |
| Survival status | Death | 166 | 31.32% |
| | Alive | 364 | 68.68% |

the expression of *AURKB* in 72 pairs of ccRCC tissues and matched non-cancerous adjacent tissues using Wilcoxon singed-rank test, and found that *AURKB* was significantly overexpressed in ccRCC tissues ($p < 0.05$) (Fig. 1B). These results suggested that *AURKB* may be a carcinogenic gene in ccRCC.

## Correlations between *AURKB* expression and clinical parameters in ccRCC patients

The relationship between *AURKB* expression and patients' clinical parameters was analyzed by R software. Results indicated that with the increase of AURKB expression, these clinical parameters (histological grade, pathological stage, T stage, N stage and M stage) also
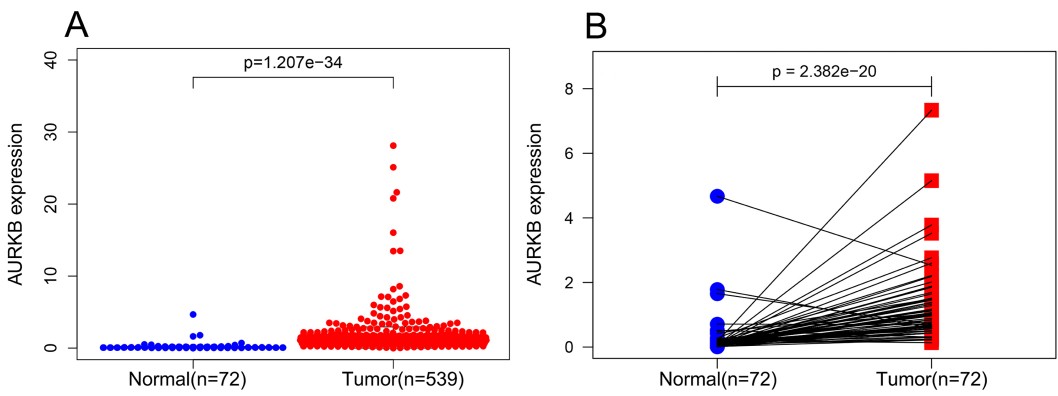

**Figure 1** *AURKB* was significantly overexpressed in ccRCC compared to normal or adjacent normal tissues. (A) *AURKB* was significantly upregulated in cancer tissues compared to normal tissues ($p < 0.05$). (B) *AURKB* was expressed at higher levels in ccRCC ($p < 0.05$) compared to 72 pairs of non-cancerous adjacent tissues.

elevated (all $p <0.05$). Furthermore, expression of AURKB in male was higher than female ($p <0.05$) (Fig. 2).

Logistic regression analysis shown that increased *AURKB* expression in ccRCC was obviously correlated with gender (OR = 1.49 for Female vs. Male, $p = 0.029$), histological grade (OR = 2.44 for G1/G2 vs. G3/G4, $p = 6.97E-07$), pathological stage (OR = 1.17 for stage I vs. stage III, $p = 0.000$; OR = 3.62 for stage I vs. stage IV, $p = 2.42683E-06$ ), TNM stage (OR = 2.84 for T1/T2 vs. T3/T4, $p = 3.12E-08$; OR = 4.75 for N0 vs. N1, $p = 0.017$; OR = 2.99, for M0 vs. M1, $p = 4.8776E-05$) (Table 2). These results indicated that ccRCC with increased *AURKB* expression is prone to progress to a more advanced stage, lymph node metastasis distant metastasis.

## Prognostic role of *AURKB* expression in ccRCC Patients

To further understand the prognostic role of *AURKB* expression in ccRCC, all ccRCC patients were categorized according to the median *AURKB* expression value (high *AURKB* expression group and low *AURKB* expression group). Patients who lacked complete clinical data were excluded from the analysis. Kaplan–Meier survival analysis indicated that the high *AURKB* expression group had worse prognosis compared with the low *AURKB* expression group ($p <0.05$) (Fig. 3). The univariate analysis indicated that high *AURKB* expression was associated with poorer OS and DFS ($p <0.05$). Other clinical parameters, such as pathological stage and histological grade, also correlated with worse OS and DFS ($p <0.05$) (Table 3).

To confirm the prognostic value of *AURKB* expression, multivariate analysis was performed. The results showed that age, histological grade, pathological stage and *AURKB* expression were independently associated with OS ($p <0.05$), and histological grade, pathological stage and *AURKB* were independently correlated with DFS ($p <0.05$). Overall,
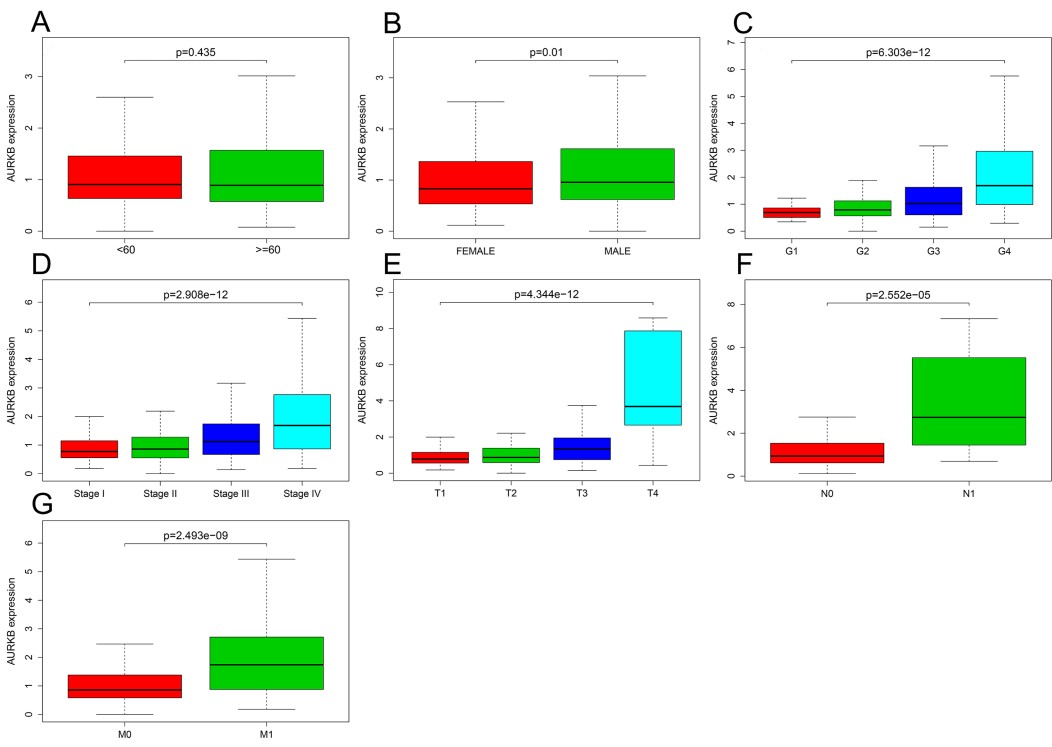

**Figure 2  Association of *AURKB* expression with clinical parameters.** (A) Age ($p > 0.05$); (B) gender ($p < 0.05$); (C) histological grade ($p < 0.05$); (D) pathological stage ($p < 0.05$); (E) T stage ($p < 0.05$); (F) N stage ($p < 0.05$); (G) M stage ($p < 0.05$).

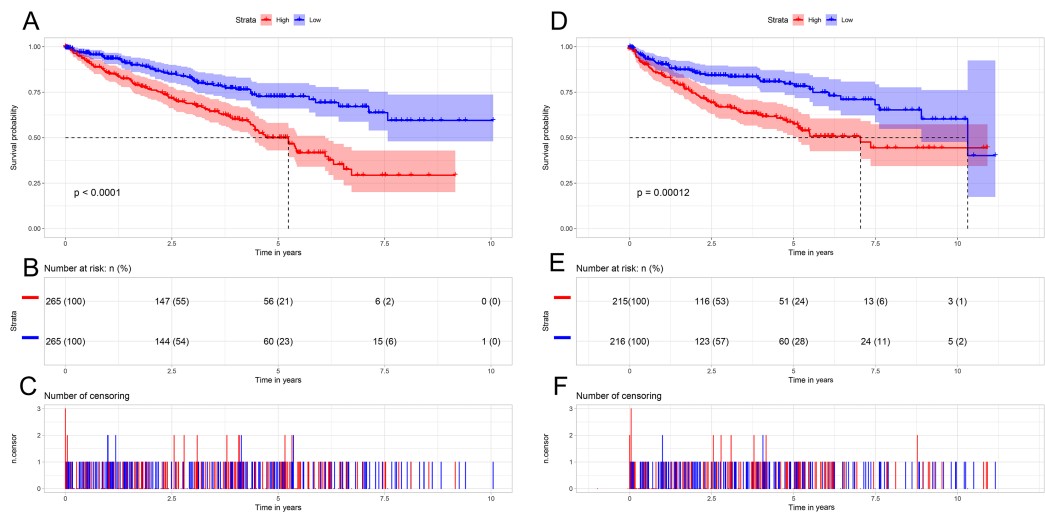

**Figure 3  Kaplan-Meier survival curves of patients with ccRCC based on *AURKB* expression levels.** (A) The Kaplan-Meier curves, (B) number at risk, and (C) number of censoring of OS in ccRCC. (D) The Kaplan-Meier curves, (E) number at risk, and (F) number of censoring of DFS in ccRCC. High expression of *AURKB* was correlated with poorer OS and DFS in ccRCC patients.

**Table 2  AURKB expression correlated with clinical parameters (logistic regression).**

| Clinical parameters | Total (N) | Odds ratio in *AURKB* expression | *p*-Value |
|---|---|---|---|
| Age | | | |
| <60 vs. ≥60 | 530 | 0.92(0.65–1.30) | 0.6631356 |
| Gender | | | |
| Female vs. Male | 530 | 1.49(1.04–2.13) | 0.02927836 |
| Histological grade | | | |
| G1/G2 vs. G3/G4 | 523 | 2.44(1.72–3.48) | 6.97E−07 |
| Pathological stage | | | |
| Stage I vs. Stage II | 326 | 1.17(0.65–2.08) | 0.5907123 |
| Stage I vs. Stage III | 394 | 2.26(1.46–3.52) | 0.000242585 |
| Stage I vs. Stage IV | 352 | 3.62(2.14–6.28) | 2.42683E−06 |
| T stage | | | |
| T1/T2 vs. T3/T4 | 530 | 2.84(1.97–4.14) | 3.12E−08 |
| Lymph nodes (N stage) | | | |
| N0 vs. N1 | 255 | 4.75(1.48–21.10) | 0.01705434 |
| Distant metastasis (M stage) | | | |
| M0 vs. M1 | 498 | 2.99(1.78–5.17) | 4.8776E−05 |

**Table 3  Univariate and multivariate cox regression analyses for OS and DFS in ccRCC patients.**

| Variables | Univariate analysis | | Multivariate analysis | |
|---|---|---|---|---|
| | HR (95% CI) | *p*-value | HR (95% CI) | *p*-value |
| Overall survival | | | | |
|    Age | 1.03(1.01–1.04) | 0.00000229 | 1.03(1.02–1.05) | 0.00000107 |
|    Gender | 0.93(0.67–1.28) | 0.662936583 | | |
| Histological grade | 2.29(1.85–2.83) | 1.94E−14 | 1.40(1.10–1.79) | 0.005937457 |
| Pathological stage | 1.88(1.64–2.16) | 4.67E−20 | 1.60(1.03–2.48) | 0.034897661 |
|    T stage | 1.94(1.63–2.29) | 1.5E−14 | 0.87(0.58–1.31) | 0.528085268 |
|    M stage | 4.28(3.10–5.90) | 7.45E−19 | 1.45(0.74–2.84) | 0.27403342 |
|    *AURKB* | 1.13(1.09–1.16) | 2.42E−13 | 1.09(1.05–1.14) | 0.00000276 |
| Disease free survival | | | | |
|    Age | 1.00(0.99–1.02) | 0.20947696 | | |
|    Gender | 1.35(0.91–2.01) | 0.132201537 | | |
| Histological grade | 2.97(2.29–3.83) | 8.11792E−17 | 1.73(1.33–2.25) | 4.49123E-05 |
| Pathological stage | 2.65(2.22–3.17) | 5.19396E−27 | 2.33(1.30–4.18) | 0.004223284 |
|    T stage | 2.5(2.02–3.08) | 1.92212E−17 | 0.79(0.46–1.33) | 0.384923805 |
|    M stage | 8.52(5.87–12.37) | 1.95879E−29 | 1.47(0.63–3.43) | 0.361781114 |
|    *AURKB* | 1.13(1.09–1.18) | 3.0124E−09 | 1.07(1.01–1.13) | 0.019973352 |

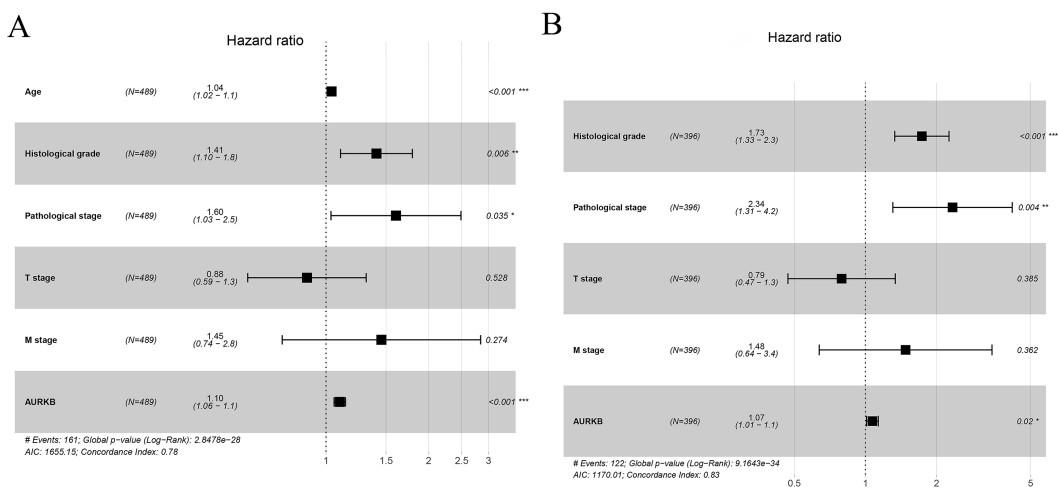

**Figure 4** **Forest plots for multivariate cox regression analyses.** (A) Age, histological grade, pathological stage and *AURKB* are independently correlated with OS; (B) Histological grade, pathological stage and *AURKB* are independently associated with DFS.

these results suggest that *AURKB* is an independent prognostic factor of ccRCC (Fig. 4 and Table 3).

## Prognostic models of *AURKB* expression and new nomograms

As the expression of *AURKB* plays an important role in OS and DFS in ccRCC, we attempted to explore whether it can be used to create better prognostic models. Two new nomograms were constructed to predict OS and DFS, 3 and 5 years after surgery (Figs. 5A and 5C). A ROC curve was used to estimate the accuracy of the two models, and the results indicated that the two models were able to accurately predict OS and DFS at 3 and 5 years after surgery (the areas under the ROC curve were 0.792 (3-year OS), 0.748 (5-year OS), 0.851 (3-year DFS) and 0.837 (5-year DFS)) (Figs. 5B and 5D).

## Sensitivity analysis of AURKB targeted drug

Cabozantinib (targets: *VEGFR*, *MET*, *RET*, *KIT*, *FLT1*, *FLT3*, *FLT4*, *TIE2*, *AXL*) and Axitinib (targets: *PDGFR*, *KIT*, *VEGFR*) are common target drug for ccRCC. Genentech Cpd 10 (targets: *AURKA*, *AURKB*) also is a target drug. To further validate *AURKB* might become a potential treatment target in ccRCC, drug sensitivity analysis of ccRCC cell lines was performed. Results showed that 2 ccRCC cell lines ($n = 31$) were sensitive to Cabozantinib, 21 ccRCC cell lines ($n = 31$) were sensitive to Genentech Cpd 10, and none ccRCC cell lines ($n = 24$) are sensitive to Axitinib (Table 4 and Fig. 6). These results suggested that ccRCC cell lines were more sensitive to Genentech Cpd 10 than Cabozantinib and Axitinib ($p < 0.05$), and *AURKB* probably become a promising target to treat ccRCC.

## Identification of *AURKB* related signaling pathways

GSEA was used to screen signaling pathways involved in ccRCC between low and high *AURKB* expression data set. GSEA indicated significant differences (FDR < 0.05, NOM *p*-value < 0.05) in enrichment of MSigDB Collection (c2.cp.v6.2.symbols.gmt).

Wan et al. (2019), *PeerJ*, DOI 10.7717/peerj.7718

**Table 4 Drugs sensitivity analysis of ccRCC cell lines.**

| Genentech Cpd 10 vs. Cabozantinib | | | | | Genentech Cpd 10 vs. Axitinib | | | | |
|---|---|---|---|---|---|---|---|---|---|
| Drug | Total (N) | Sensitive | Resistent | $\chi^2$ | $p$-value | Drug | Total (N) | Sensitive | Resistent | $\chi^2$ | $p$-Value |
| Genentech Cpd 10 | 31 | 21 | 10 | 22.395 | 2.22E−06 | Genentech Cpd 10 | 31 | 21 | 10 | 23.508 | 1.244E-06 |
| Cabozantinib | 31 | 2 | 29 | | | Axitinib | 24 | 0 | 24 | | |

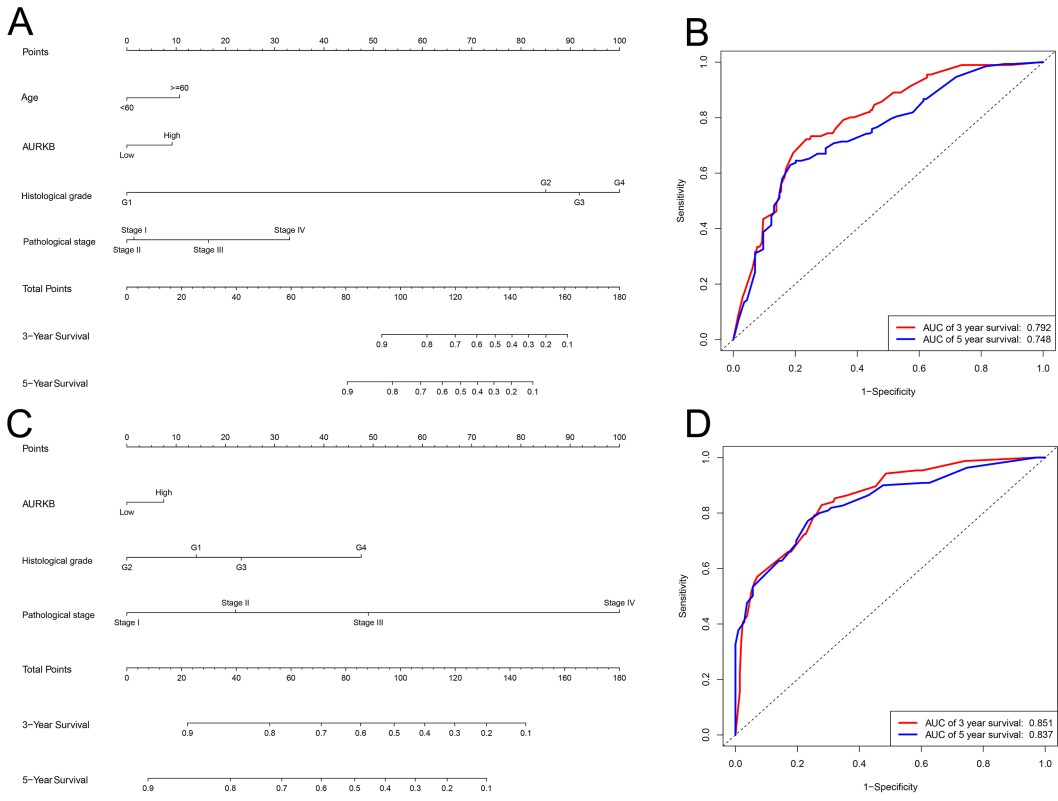

**Figure 5  Nomogram and ROC plots for the prediction of outcome in patients with ccRCC.** Nomogram for the prediction of OS (A) and PFS (B) at 3 and 5 years after surgery. ROC curve evaluated the accuracy of the two models (C and D), and the areas under the ROC curve were 0.792 (3-year OS), 0.748 (5-year OS), 0.851 (3-year DFS) and 0.837 (5-year DFS), and indicated a good accuracy.

Twenty-one signaling pathways involved in autoimmune thyroid disease, intestinal immune network for IgA production, antigen processing and presentation, cytokine-cytokine receptor interaction, asthma, type I diabetes mellitus, primary immunodeficiency, graft versus host disease, allograft rejection, base excision repair, homologous recombination, natural killer cell mediated cytotoxicity, cytosolic DNA sensing pathway, viral myocarditis, hematopoietic cell lineage, DNA replication, systemic lupus erythematosus, leishmania infection, cell cycle, cell adhesion molecules (CAMs) and proteasome were differentially enriched in the *AURKB* high expression phenotype (Table 5). Five signaling pathways that may be closely connected to the progression of ccRCC tumors are shown in Fig. 7.

## DISCUSSION

Many studies have suggested that *AURKB* plays a vital role in tumorigenesis and tumor progression (*Zhu et al., 2019*; *Mesic et al., 2017*; *Kotian et al., 2017*). *AURKB* has been shown to be involved in the development of breast cancer, and its expression has been associated with breast cancer prognosis (*Liao et al., 2018*; *Naorem, Muthaiyan &*

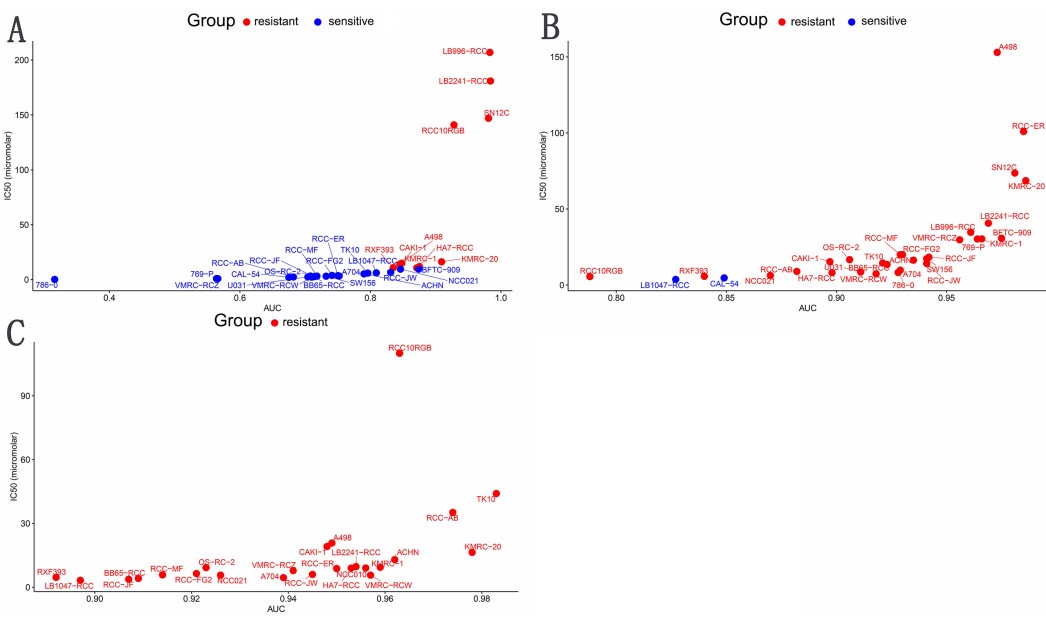

**Figure 6** **Drugs sensitivity analysis of ccRCC cell lines.** (A) Genentech Cpd 10, (B) Cabozantinib, (C) Axitinib. Those cell lines whose IC50 value greater than max screening concentration was regarded as resistant to target drugs. Blue represents sensitive ccRCC cell lines to target drugs; red indicates resistant ccRCC cell lines to target drugs; IC50, natural log half maximal inhibitory concentration; AUC, Area under the dose-response curve.

*Venkatesan, 2019*). In non-small cell lung cancer (NSCLC), *AURKB* has been shown to be overexpressed and correlated with poorer prognosis of patients. Its overexpression was shown to significantly promote proliferation of NSCLC cells via inhibiting the p53-related pathway. In addition, expression of *AURKB* has also been associated with drug resistance in NSCLC. Overexpression of *AURKB* increased drug resistance in NSCLC cells, whereas *AURKB* knockdown re-sensitized NSCLC cells to chemotherapeutic drugs (*Yu et al., 2018*). In colorectal cancer (CRC), *AURKB* has also been shown to act as an important oncogenic factor, be involved in the development of CRC, and promoted drug resistance and progression of CRC though regulation of the Wnt signaling pathway and the p53-related pathway (*Subramaniyan, Kumar & Mathan, 2017*; *Nair et al., 2009*; *Wu et al., 2011*; *Pohl et al., 2011*). All these studies have suggested that *AURKB* promotes carcinogenesis and is associated with drug resistance.

In this work, we sought to identify the role of *AURKB* expression in ccRCC progression, particularly, its role as a prognostic factor in ccRCC. Moreover, we also attempted to screen *AURKB*-related signaling pathways in ccRCC to contribute to the understanding the potential mechanism involved in the regulation of ccRCC development by *AURKB*.

Firstly, we compared that expression of *AURKB* in ccRCC and normal tissues. The results showed that *AURKB* was overexpressed in ccRCC tissues compared to normal tissues, and its expression was associated with pathological stage, histological grade, T stage, M stage, and N stage.

**Table 5  Gene sets enriched in the high *AURKB* expression phenotype.**

| Gene set name | NES | NOM *p*-value | FDR *q*-value |
|---|---|---|---|
| KEGG_AUTOIMMUNE_THYROID_DISEASE | 2.3979816 | 0 | 0.00331709 |
| KEGG_INTESTINAL_IMMUNE_NETWORK_FOR_IGA_PRODUCTION | 2.3240876 | 0 | 0.004554952 |
| KEGG_ANTIGEN_PROCESSING_AND_PRESENTATION | 2.2065759 | 0.007889546 | 0.012016102 |
| KEGG_CYTOKINE_CYTOKINE_RECEPTOR_INTERACTION | 2.1696072 | 0.001915709 | 0.013810257 |
| KEGG_ASTHMA | 2.1645305 | 0 | 0.011436771 |
| KEGG_TYPE_I_DIABETES_MELLITUS | 2.139911 | 0 | 0.012462393 |
| KEGG_PRIMARY_IMMUNODEFICIENCY | 2.0994904 | 0 | 0.014886827 |
| KEGG_GRAFT_VERSUS_HOST_DISEASE | 2.0993276 | 0 | 0.013095205 |
| KEGG_ALLOGRAFT_REJECTION | 2.0911534 | 0 | 0.012559181 |
| KEGG_BASE_EXCISION_REPAIR | 2.047824 | 0.001901141 | 0.016551593 |
| KEGG_HOMOLOGOUS_RECOMBINATION | 2.047304 | 0.003913894 | 0.015096117 |
| KEGG_NATURAL_KILLER_CELL_MEDIATED_CYTOTOXICITY | 2.0085354 | 0.015655577 | 0.019628806 |
| KEGG_CYTOSOLIC_DNA_SENSING_PATHWAY | 2.00772 | 0 | 0.018204125 |
| KEGG_VIRAL_MYOCARDITIS | 1.993407 | 0.009940358 | 0.019238696 |
| KEGG_HEMATOPOIETIC_CELL_LINEAGE | 1.9526478 | 0.005825243 | 0.024869524 |
| KEGG_DNA_REPLICATION | 1.9470055 | 0.013461539 | 0.024576483 |
| KEGG_SYSTEMIC_LUPUS_ERYTHEMATOSUS | 1.9417096 | 0.013944224 | 0.02391057 |
| KEGG_LEISHMANIA_INFECTION | 1.9209716 | 0.027559055 | 0.026700974 |
| KEGG_CELL_CYCLE | 1.8969755 | 0.025590552 | 0.030597683 |
| KEGG_CELL_ADHESION_MOLECULES_CAMS | 1.8593365 | 0.03206413 | 0.037772134 |
| KEGG_PROTEASOME | 1.8296527 | 0.028248588 | 0.044589847 |

**Notes.**

NES, normalized enrichment score; NOM, nominal; FDR, false discovery rate.

Gene sets with NOM *p*-value <0.05 and FDR *q*-value <0.05 were regarded as significantly enriched.

Secondly, Kaplan–Meier survival analysis showed that compared to the low *AURKB* expression group, the high *AURKB* expression group of patients had poorer OS and DFS. Moreover, some variables were also associated with the prognosis of ccRCC patients, including pathological stage, histological grade, T stage, and M stage. In addition, multivariate analysis confirmed that *AURKB* expression was a prognostic factor. Another, the drug sensitivity analysis of ccRCC cell lines suggested that various cell lines were sensitive to Genentech Cpd 10, and AURKB might be a promising target to treat ccRCC.

Finally, we constructed prognostic models of *AURKB* expression, and the area under the curve (AUC) values proved that the new prognostic models can accurately predict OS and PFS. Furthermore, *AURKB* related signaling pathways in ccRCC were analyzed by GSEA, and results suggested that intestinal immune network for IgA production, cytokine-cytokine receptor interaction, natural killer cell mediated cytotoxicity, cell cycle and cell adhesion molecules (CAMs), correlate with progression of ccRCC. It has been shown that intestinal immune network for IgA production play a pivotal role in tumor progression (*Liang et al., 2018*). *Yang et al. (2018)* have found that activation of intestinal immune network for IgA production signaling pathway promoted malignant behavior of tumor cells. Additionally, cytokine-cytokine receptor interaction was a significant immune signaling pathway, as it can modulate interaction of cytokines, thereby regulating occurrence and progression of

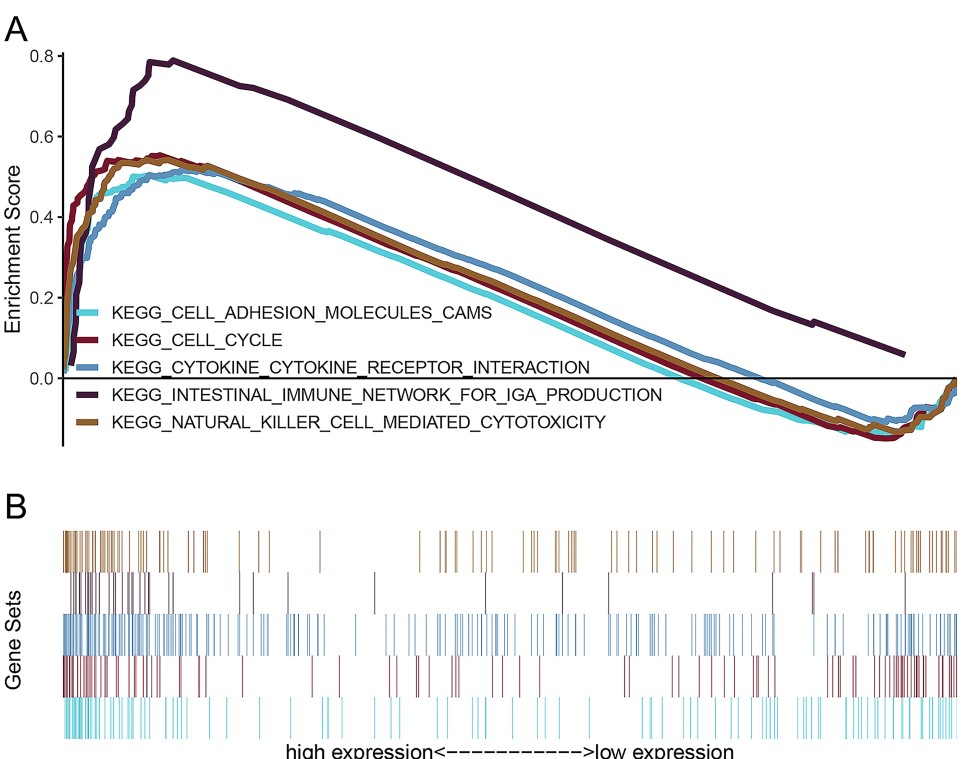

**Figure 7** **Enrichment plots from gene set enrichment analysis (GSEA).** (A) Enrichment sore, (B) gene sets. Intestinal immune network for IgA production, cytokine-cytokine receptor interaction, natural killer cell mediated cytotoxicity, cell cycle, and cell adhesion molecules (CAMs) might be highly correlated with progression of ccRCC.

cancers (*Tumino et al., 2019*; *Nagarsheth, Wicha & Zou, 2017*). Natural killer cell mediated cytotoxicity plays a vital role in modulating tumor microenvironment; their activation was closely related to the progression of tumors and prognosis of cancer patients (*Malmberg et al., 2017*; *Chan, Wucherpfennig & De Andrade, 2019*; *Bassani et al., 2019*). The cell cycle controls the progression of tumors (*Roy et al., 2017*), and its activation can significantly promote proliferation of tumor cells, thus accelerating tumor growth (*Eifler & Vertegaal, 2015*; *Mast et al., 2019*). Increasing amount of evidence has indicated that alterations in the adhesion properties of neoplastic cells play an important role in the development and progression of tumors, and cell adhesion molecules (CAMs) are involved in the adhesion of neoplastic cells, and participate in metastasis, migration and invasion of tumors (*De Méndez & Bosch, 2011*; *Xin, Dong & Guo, 2015*; *Okegawa et al., 2004*). All these results suggest that *AURKB* promotes oncogenesis and progression of ccRCC through regulating multiple signaling pathways.

At present, many studies have already indicated that *AURKB* is a promising therapeutic target in various cancers, such as non-small cell lung cancer (NSCLC) (*Bertran-Alamillo et al., 2019*), gastric cancer (GC) (*He et al., 2019*), leukemia (*He et al., 2016*), prostate cancer (PC) (*Addepalli et al., 2010*), and breast cancer (*Han et al., 2017*). Additionally, in the

present study, we found that *AURKB* is a promising biomarker in the treatment of ccRCC and a predictor of prognosis.

Inevitably, our study also has several limitations. Firstly, the data we analyzed in the present study were extracted from several public databases, which had not been verified. Secondly, the mechanisms by which AURKB regulates the occurrence and progression of ccRCC need further exploration. Finally, our study found that ccRCC cell lines were more sensitive to AURKB-targeting drug (Genentech Cpd 10) than the conventional targeted drugs (Cabozantinib and Axitinib). However, more studies are necessary to identify whether AURKB could be used as a target for ccRCC treatment.

## CONCLUSIONS

In summary, our study suggests that *AURKB* is over-expressed in ccRCC, and it is a valuable prognostic factor for predicting OS and DFS of ccRCC patients. *AURKB* can promote development of ccRCC via various signaling pathways including intestinal immune network for IgA production, cytokine-cytokine receptor interaction, natural killer cell mediated cytotoxicity, cell cycle and cell adhesion molecules (CAMs). In addition, *AURKB* might be a promising therapeutic target for ccRCC. However, more research is required to verify the findings of this study.

### Funding
The study was supported by the Natural Science Foundation of Hainan Province (No. 819MS136). The funders had no role in study design, data collection and analysis, decision to publish, or preparation of the manuscript.

### Grant Disclosures
The following grant information was disclosed by the authors:
Natural Science Foundation of Hainan Province: 819MS136.

### Competing Interests
The authors declare there are no competing interests.

### Author Contributions

- Bangbei Wan conceived and designed the experiments, performed the experiments, analyzed the data, contributed reagents/materials/analysis tools, prepared figures and/or tables, authored or reviewed drafts of the paper, approved the final draft.
- Yuan Huang conceived and designed the experiments, performed the experiments, analyzed the data, authored or reviewed drafts of the paper, approved the final draft.
- Bo Liu and Likui Lu performed the experiments, analyzed the data, contributed reagents/materials/analysis tools, prepared figures and/or tables.
- Cai Lv conceived and designed the experiments, analyzed the data, prepared figures and/or tables, authored or reviewed drafts of the paper, approved the final draft.

## Data Availability

The raw measurements are available in the Supplemental Files.

## Supplemental Information

Supplemental information for this article can be found online at http://dx.doi.org/10.7717/peerj.7718#supplemental-information.

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
