# Peer review of "AURKB: a promising biomarker in clear cell renal cell carcinoma"

_PeerJ, doi:10.7717/peerj.7718_

## Round 0.1 · original submission · Major Revisions

The reviewers have raised important queries on clarity and content. A few minor but important questions on the validity of the findings have been raised.

·

Basic reporting

Introduction

I suggest writing a more detailed introduction; in particular you need to write more about what is known about AURKB. You mention that AURKB expression is related to tumorigenesis and tumor progression (lines 51 - 53) but you need to provide more details about these studies.

Introduction sounds more like an abstract than introduction. It is very short and it needs a lot of work.

Results

Results need to be better explained providing the reader with more details of what do you think the data means. Lines 96-101 – you need to explain in more details. Similarly, lines 103-106. These are not fully explained. How is AURKB correlated with gender, histological grade, pathological stage, T stage etc? This is not explained at all and is poorly written.

Tables
Please name Tables – Table not Tbale
Table 2 – Please put a space after Total (N)

Figures
Figure 2- Please enlarge its size a little.
Figures 3, 4 and 5 are very small to see. Please enlarge their size.

Experimental design

Methods need to be described in more details. Research question is well defined but the authors need to provide the reader with more details.

Validity of the findings

The data need to be better presented and explained. Some figures are too small (See the comments below). Conclusions are stated but very briefly.

Additional comments

I suggest the author to work on this article because Introduction and Results need a lot of work.

Reviewer 2 ·

Basic reporting

English used throughout the manuscript is acceptable and within the scientific style.
Literature review and background information is NOT sufficient enough to convince why the study was carried.
The study is NOT hypothesis driven.
The results could have been further elaborated.

Experimental design

The study was completely bioinformatics based, therefore analysis tool was satisfactory to meet the level of study. However, the study was not hypothesis driven. It was not clear why authors chose to study levels of Aurora B kinase in clear cell renal cell carcinoma. To me it seems it is very premature to conclude that Aurora Kinase B may be the promising biomarker for the treatment of ccRCC, esp. because Aurora Kinase B is a kinase and it has several downstream targets that may have direct effect on the proliferation of tumor cells. Therefore, without detail mechanistic studies, it can not be concluded Aurora B as a molecular marker.
Investigative and statistical tools used were satisfactory.
Since this study is a purely bioinformatics based, the method provided is sufficient enough.

Validity of the findings

Authors examined the mRNA expression levels of Aurora Kinase B in clear cell Renal cell carcinoma. Furthermore, authors studied the correlation between expression levels of Aurora Kinase B and disease progression. Depending on the scope of the journal and reader group, the data presented can be sufficient enough to be considered to be published in a cancer focussed journal with few revisions. However, this study is not suitable for the journals and readers who are focussed on understanding the fundamental mechanism of Aurora B biology.
Authors had performed detailed bioinformatics based analysis to conclude that Aurora Kinase B may be a promising biomarker for the treatment of clear cell Renal Cell Carcinoma, however, this conclusion is too premature. Authors need to back up the data with some laboratory based findings in cell line or perform some orthogonal validation using cell line databases such as CCLE.

Additional comments

1) Logical basis for studying expression levels of Aurora B kinase and its correlation with the disease prognosis in clear cell Renal Cell Carcinoma is missing. Authors need to do more literature reviews, not only the cancer studies, but also the basis science studies on Aurora B.
2) Overexpression or underexpression of Aurora kinase B is known to have effect on chromosome instability (CIN). What is the ploidy status of ccRCC? It is possible that increased Aurora B expression and positive correlation with poor prognosis could be due to aneuploid nature of the tumors. Please use an appropriate database to examine the CIN in these tumors and whether there is any correlation with Aurora B levels and prognosis.
3) For further detail understanding on effect of Aurora B levels in ccRCC and its prognosis, survival rate etc., it is important to study the drugs efficiency in these cancers? Any drug resistance correlates with Aurora B activity?
4) It is interesting to study the synthetic dosage lethality partners for Aurora B overexpression in ccRCC. This finding may predict the therapeutic target for ccRCC. The data from cancer database may be orthogonally validated with cell line database studies.

---

## Round 0.2 · Minor Revisions

Authors have tried to address most queries. The manuscript has improved in quality.

Reviewer 2 raises valid concerns - they could either address these by experiments or discuss the lack of these data by text and tone down their claims. Point-1 of Reviewer-2 is particularly important to address.

·

Basic reporting

The authors have improved the paper. The use of english language is acceptable. Introduction is well written and details have been added. The figures, tables, data are shared. The results and discussion are well explained.

Experimental design

Research questions have been well defined. The authors state clearly how their findings fill an identified knowledge gap. The investigation has been well carried. Methods are described with sufficient details.

Validity of the findings

The literature is well written. All underlying data have been provided and explained in details. Conclusions are well stated, linked to the original question and clearly support the results.

Additional comments

This is a very interesting paper. It clearly explains how AURKB can be used as a promising biomarker in clear cell renal cell carcinoma.

Reviewer 2 ·

Basic reporting

In the rebuttal letter, authors explained that they examined the expression levels of Aurora B kinase in ccRCC because other studies had shown that Aurora B kinase levels are higher in gastric cancer, thyroid cancer etc. This is not a sufficient and logical explanation for studying Aurora B levels in ccRCC. Authors have included more information on Aurora B levels in different cancers and non-neoplastic diseases. However, these information is not relevant to what they are reporting in this study.

Experimental design

Experimental method section has been modified significantly.

Validity of the findings

1) Authors added new results on the effects of drugs on ccRCC cell lines and showed that these cell lines are sensitive to Aurora B kinase targeting drugs, Genetech Cpd10. This result tells that inhibiting Aurora B kinase is lethal for ccRCC cells. Since Aurora B kinase is an important mitotic kinase, its function is important for cell viability and inhibiting Aurora B kinase causes global cell death. To rule this out authors should have examined the effects of Genetech Cpd10 on non-ccRCC cell lines as they did for the effects of Axitinib in non-ccRCC cell lines. Therefore, I am not convinced that Aurora B targeting drugs are sensitive to Aurora B high expressing cells only.
2) If authors want to make point that Aurora B targeting drugs are sensitive to ccRCC cell lines, they should back up this data by examining the OS of ccRCC patients who are on Aurora B targeting drugs and show that they have better OS compared to those on non-Aurora B targeting drugs.
3) I understand that the authors could not find the appropriate database to examine the ploidy status of ccRCC.
4) Synthetic dosage lethality (SDL) is not related to the drugs sensitivity. SDL genes are those that are always upregulated or downregulated/mutated with the Aurora B overexpression.

Additional comments

Overall, authors have revised the manuscript significantly, however I still think that there are several results that need validation.

---

## Round 0.3 · accepted · Accept

The recent text edits by the authors presents a balanced discussion of their observations and addresses the queries raised by reviewers.